# Discovery and SAR Study of Quinoxaline–Arylfuran Derivatives as a New Class of Antitumor Agents

**DOI:** 10.3390/pharmaceutics14112420

**Published:** 2022-11-09

**Authors:** Dongmei Fan, Pingxian Liu, Yunhan Jiang, Xinlian He, Lidan Zhang, Lijiao Wang, Tao Yang

**Affiliations:** 1Laboratory of Human Diseases and Immunotherapies, West China Hospital, Sichuan University, Chengdu 610041, China; 2Institute of Immunology and Inflammation, Frontiers Science Center for Disease-Related Molecular Network, West China Hospital, Sichuan University, Chengdu 610041, China; 3Department of Cardiovascular Surgery, West China Hospital, Sichuan University, Chengdu 610041, China; 4State Key Laboratory of Biotherapy and Cancer Center, West China Hospital, Sichuan University, Chengdu 610041, China; 5School of Food and Bioengineering, Xihua University, Chengdu 610041, China

**Keywords:** antitumor agent, quinoxaline–arylfuran, ROS, drug discovery

## Abstract

A novel class of quinoxaline–arylfuran derivatives were designed, synthesized, and preliminarily evaluated for their antiproliferative activities in vitro against several cancer cell lines and normal cells. The representative derivative **QW12** exerts a potent antiproliferative effect against HeLa cells (IC_50_ value of 10.58 μM), through inducing apoptosis and triggering ROS generation and the accumulation of HeLa cells in vitro. Western blot analysis showed that **QW12** inhibits STAT3 phosphorylation (Y705) in a dose-dependent manner. The BLI experiment directly demonstrated that **QW12** binds to the STAT3 recombination protein with a KD value of 67.3 μM. Furthermore, molecular docking investigation showed that **QW12** specifically occupies the pY+1 and pY-X subpocket of the SH2 domain, thus blocking the whole transmission signaling process. In general, these findings indicated that the study of new quinoxaline–aryfuran derivatives as inhibitors of STAT3 may lead to new therapeutic medical applications for cancer in the future.

## 1. Introduction

Today, cancer is among the leading causes of mortality and a major public health concern in the world. Approximately 19.3 million cases occurred in 2020 alone [1]. If current trends continue, new cancer cases will increase by 60% worldwide by 2040 and by more than 80% in low- and middle-income countries, where cancer diagnostic and treatment services are insufficient [1]. Although the increased understanding of tumorigenesis and progression has led to the development of numerous drugs available to treat various types of cancer, the marketing of anticancer drugs could not meet the demand of clinics until now [2]. Therefore, it is paramount to discover novel drugs for cancer therapies in order to minimize patient suffering and reduce the costs of expensive treatments.

Quinoxalines and their derivatives have been considered promising active compounds for the development of novel therapeutic agents, because of their broad biological activities, such as anticancer [3], anti-inflammatory [4], antifungal [5], antiproliferative [6], and antitubercular [7]. Many anticancer candidates with quinoxaline scaffolds (Figure 1) have been subjected to clinical trials, such as antineoplastic topoisomerase II inhibitors XK469 (**1**) and CQS (**2**) [8]. Compound **3**, a synthesized derivative of acrylamide–quinoxaline, with moderate inhibitory activity on tumor cell lines derived from patients resistant to a first-generation EGFR inhibitor [9]. Compound **4** exhibited potent cytotoxicity against cancer cells and significantly reduced tumor size in a dose-dependent manner [10]. Compound **5**, a new quinoxaline–isoselenourea hybrid, showed good activity against five melanoma cell lines, including mutant B-RAFV600E and wild-type, with IC_50_ values of 0.8–3.8 μΜ [11]. NVP-BSK805 (**6**), a new substituted quinoxaline derivative, exhibited inhibition against six human myeloma cell lines, with IC_50_ values between 2.6 µM and 6.8 µM [12]. Compound **7** (Figure 2), a phenyloxy quinoxaline derivative, which potently suppresses proliferation against SET-2 cells in vitro and favorable rat pharmacokinetic properties in vivo [13]. This accumulated evidence indicated that quinoxaline was an important crucial heterocycle in the development of antitumor agents.

In addition to the quinoxaline moiety, the furan ring is always widely used in drug development for its broad bioactivities. For example, nifuroxazide exhibits potent antiproliferative activity against various melanoma cell lines by inhibiting STAT3, which could significantly inhibit tumor growth without obvious side effects in a model of A375-bearing mice, by inducing apoptosis and reducing cell proliferation and metastasis [14,15,16]. Furthermore, certain furan derivatives with phenyloxy substituents, such as compounds **8** and **9** (Figure 2), have been obtained and exhibited high cytotoxicity against the human cancer cell lines MCF-7, TK-10, and UACC-62 [17]. It has been reported that phenyl-2-furan is a potential pharmacophore for antitumor proliferative activity with multiple mechanisms, such as inhibiting protein tyrosine phosphatase 1B [18] and inhibiting P-glycoprotein [19]. Thus, it is promising to use the phenyl-2-furan motif to discover anticancer agents.

As an ongoing effort to identify effective anticancer agents with novel scaffolds, we combined quinoxaline and arylfuran motifs as the designed quinoxaline–arylfuran scaffold (Figure 2). Recently, substituted hydrazide–hydrozones have attracted the researchers to develop their heterocyclic analogues as medicinal agents for their diverse biological activities, such as antibacterial [20], antitumor [21], and anti-inflammatory [22]. Thus, we chemically constructed a quinoxaline–arylfuran scaffold with a substituted acylhydrazone side chain (Figure 2) and evaluated it for its anticancer properties against several cancer cell lines. The preliminary structure–activity relationship was studied, and then we performed cell cycle analysis and apoptosis studies on the most potent compound, **QW12**, in vitro. Furthermore, Western blot and BLI analysis revealed the mechanism of action of this compound. These results indicated that **QW12** could be further investigated as an ideal lead compound for antitumor agents.

## 2. Materials and Methods

### 2.1. Synthetic Procedures and Analytical Data

All reagents and chemicals were commercially available and used without further purification. ^1^H and ^13^C NMR spectra were recorded on a Bruker Avance 300 spectrometer (400 and 101 MHz for ^1^H and ^13^C NMR, respectively) in CDCl_3_ or DMSO-*d*_6_. MS spectra were recorded using an Agilent spectrometer (9575c inert MSD; Agilent Technologies, Santa Clara, CA, USA).

Detailed chemical synthesis procedures and chemical analysis results of intermediates (**11a**–**c**, **12a**–**c**, and **14a**–**c**) and final compounds **QW1**–**24** are described in Appendix A.

### 2.2. Cell Culture

Human lung carcinoma cell lines A549, human cervical cancer cell lines HeLa, human prostate cancer cell lines PC3, human colorectal carcinoma cell lines HCT116, and human liver cells L02 were purchased from the Chinese Academy of Sciences Shanghai Cell Bank (Shanghai, China). HeLa, A549, PC3, HCT116, and L02 cells were cultured in DMEM Dulbecco’s Modified Eagle Medium (DMEM; Gibco Inc., Gaithersburg, MD, USA). All cultures were provided with 10% fetal bovine serum (FBS; YHSM, Beijing, China), 100 IU/mL penicillin, and 100 μg/mL streptomycin, and cells were incubated at 37 °C in an atmosphere of 5% CO_2_.

### 2.3. CCK-8 Assay

Cell viability was determined using a CCK-8 kit (Apexbio, Houston, TX, USA). Cells were seeded at the number of 9000 per well in a 96-well plate for 12 h and then treated with **QW12** for 24 h before receiving treatment with 10 μL CCK-8 reagent treatment for 1–4 h. Subsequently, absorbance was assessed at 450 nm. Three duplicates were set in each group.

### 2.4. Hoechst-33342 Staining

HeLa cells (4 × 10^5^ cells) were incubated in a 6-well plate for 12 h. The cells were then treated with **QW12** at the indicated concentrations for 24 h. The cells were washed twice with PBS. The cells were then incubated with Hoechst-33342 (10 μg/mL) for 5 min at room temperature in darkness. After incubation, stained cells were observed under an inverted phase contrast microscope (OLYMPUS IX73, Tokyo, Japan).

### 2.5. Immunoblotting

HeLa cells were seeded in a 6-well plate overnight and treated with **QW12** for 24 h. The cells were then harvested and washed with cold PBS and lysed with RIPA buffer supplemented with a protease and phosphatase inhibitor cocktail. Protein concentration was measured using a BCA protein assay kit (Boster Biological Technology Co., Ltd., Wuhan, China). Proteins were separated by 10% SDS-PAGE gel and transferred onto polyvinylidene difluoride (PVDF) membranes (Millipore Co., Billerica, MA, USA). After blocking with 5% mild in PBST for 1 h at room temperature, membranes were incubated with specific primary antibodies against STAT3 (#ET1605-45), Bcl-2 (#ET1603-11), Bax (#ET1603-34), or GAPDH (#HA721136) overnight at 4 °C. The above antibodies were purchased from HUABIO, China. Phospho-STAT3 (Tyr 705) (#9145S) and anti-rabbit IgG, HRP-linked antibody (#7074S) were purchased from Cell Signaling Technology, Inc. America. The next day, the membranes were washed with PBST three times and incubated with secondary antibodies 2 h at room temperature. Images were captured and documented with a CCD system (Tanon 5200, Biotanon, Shanghai, China).

### 2.6. Flow Cytometry Assay

For cell cycle analysis, cells were collected after treatment, washed twice with cold PBS, fixed in 70% ethanol overnight, and stained with PI (50 mg/mL, Sigma, Shanghai, China), plus 0.2 mg/mL of DNase-free RNase A (Qiagen) for 1 h at 4 °C.

For the Annexin V apoptosis assay, cells were collected at the end of the treatment, washed with cold PBS, and stained with the PI/Annexin V-FITC apoptosis kit (4A Biotech Co., Ltd., Beijing, China) following the manufacturer’s instructions. All flow cytometric analyses were performed at the Tianfu Science and Technology Park Molecular and Cell Platform in the West China Hospital.

Intracellular reactive oxygen species (ROS) were measured by 2,7-dichlorofluorescein diacetate (DCFH-DA) staining. HeLa cells were treated with **QW12** at 10 μM and 40 μM for 6 h. After indicated treatment, cells were harvested and washed with PBS. Cells were incubated with 10 μM DCFH-DA at 37 °C for 30 min. Cells were washed twice, resuspended with PBS, and subjected to flow cytometry.

### 2.7. Wound Healing Assay

A total of 1 × 10^6^ cells/well in the logarithmic growth phase were seeded in 6-well plates. When cell density reached 80 to 90%, a scratch was made in the monolayer in the middle of the well with a 200 μL pipette tip. The tip was kept perpendicular to the bottom of the well to obtain a straight gap. The detached cells were washed away and removed. The healing of the wound within the same scraped line was then observed and photographed at the indicated time points (0 h, 24 h, 48 h). Each experiment was repeated three times.

### 2.8. Colony Formation Assay

HeLa cells were digested and seeded directly in 6-well plates (1 × 10^3^ cells/well) for the colony-formation assay and cultured in the presence of 10% FBS and 1% penicillin/streptomycin at 37 °C with 5% CO_2_ for 12 h in confluent cell monolayers. The cells were then treated with **QW12** at the indicated concentrations for 24 h. Two weeks later, the medium was removed, and the plates were washed with phosphate-buffered saline (PBS) three times. Cells were fixed with anhydrous ethanol for 30 min and then stained with Giemsa dye solution (Solarbio) for 30 min. The plates were dried with a blower to ensure high-quality images were obtained. Colonies were defined as >50 cells/colony.

### 2.9. Biolayer Interferometry (BLI) Assay

BLI assays were performed using the BLItz system (Sartorius). Biosensors were first equilibrated 10 min in 1X kinetic buffer (Sartorius) consisting of PBS with 0.02% Tween20 and 0.1% BSA. Subsequently, depending on the assay, STAT3 was loaded onto the corresponding biosensor at a concentration of 1.9 µg/µL, as indicated in the Sartorius biosensor datasheets, for an appropriate time interval. The concentration range for the associating protein was chosen, when possible, based on the KD value available from the literature or experimentally determined for all scenarios wherein KD values were unknown. The data recorded were analyzed by means of the BLItz software and MATLAB to extrapolate the kinetic parameters. All association and dissociation curves were fitted by a single exponential function. Each acquisition was repeated twice to confirm reproducibility.

### 2.10. Molecular Docking

The molecular docking studies with **QW12** were performed using Schrodinger based on the crystal structure of STAT3 downloaded from the Protein Data Bank (PDB:1BG1). Schrodinger software was used to prepare the macromolecule and ligand. The compound **QW12** was docked using the Glide XP docking procedure. Gasteiger charges were assigned to the ligands by AutoDock Tools. The grid size was set to 105 Å × 75 Å × 68 Å, which is large enough to cover the entire active target site. After completing 10 million energy evaluations, the root-mean-square-deviation threshold was established as 1.5 Å, and all conformations of the ligands in the binding pocket of the macromolecule were clustered. The lowest energy clusters were identified, and the binding energy was evaluated. PyMOL (version 2.4.1) was used to create binding poses.

## 3. Results and Discussion

### 3.1. Synthesis

As shown in Figure 1, these quinoxaline–arylfuran derivatives were synthesized according to the literatures [23,24]. Briefly, the synthesis begins with (2,5-dimethoxyphenyl) boronic acid turned into intermediates **11a**–**c**, followed by treatment with NBS in DCM to obtain bromo derivatives **12a**–**c**, which yield arylfuran derivatives **14a**–**c** through treatment with 2-formylfuran-5-boronic acid. The intermediates **14a**–**c** were reacted with hydrazide compounds to generate the corresponding hydrazone derivatives **QW1**–**24**.

### 3.2. Biological Evaluation

#### 3.2.1. Evaluation of In Vitro Antitumor Activity

All synthesized quinoxalines compounds were tested for in vitro cytotoxicity against four cancer cell lines (HeLa, PC3, A549, and HCT116). The inhibitory effects of these compounds were evaluated using the CCK-8 assay after treatment with 20 μM of each compound. As shown in Table 1, most of the compounds exerted weak inhibitory activity against HeLa, PC3, A549, and HCT116 cell lines. Apparently, the quinoxaline scaffold is well tolerated, and most of the quinoxaline derivatives (**QW1**–**16**) exhibited higher activity than the naphthalene ring substitution derivatives (**QW17**–**24**). The species of phenyl substituents have a significant effect on the activity. The hydrazide side chain is *p*-cresol or pyridine, which showed higher inhibitory activity, such as in **QW3**–**5** and **QW11**–**13**, indicating that the side chain with a hydrogen-bond acceptor or donor is more appropriate for promoting activity. This result was consistent with our molecular docking analysis in the docking-study section. By comparison, hydrophobic substitution at the phenyl ring gave derivatives with relatively low potency (**QW7**–**8**). In addition, large substituents on the phenyl ring exerted an unfavorable influence on the potency of the compounds (**QW8** and **QW16**). Among these compounds, **QW12** exhibited excellent activities against HeLa, PC3, A549, and HCT116 cancer cell lines, with inhibition rates of 76.35%, 52.55%, 50.78%, and 65.43%, respectively. The positive control we used was nifuroxazide, a STAT3 inhibitor, and its inhibition rates of HeLa, PC3, A549, and HCT116 were 80.38%, 75.29%, 91.23%, and 85.31%, respectively.

To further evaluate the anticancer effect of **QW12**, we evaluated its cytotoxicity against four human cancer cell lines: HeLa (Human cervical tumor), PC3 (Human prostate tumor), HCT116 (Human colorectal carcinomas), and A549 (Human lung carcinoma). To demonstrate whether **QW12** would show the expected selectivity between normal cells vs. cancer cells, the cytotoxicity of **QW12** against normal human hepatocytes L02 was also determined. As presented in Table 2, the IC_50_ value of **QW12** against A549 was 20.57 μM, while it showed potent cytotoxicity at low micromolar concentration (10.58–12.67 μM) against the other tumor cell lines. Furthermore, the IC_50_ values of **QW12** against PC3 and HCT116 were similar to those of nifuroxazide. Interestingly, compound **QW12** showed higher safety against normal human hepatocyte L02 cells than nifuroxazide (>50 μM vs. 25.30 μM).

#### 3.2.2. **QW12** Inhibits the Proliferation and Migration of HeLa Cells

The wound-healing assay is a simple and cost-effective way to assess the invasiveness and migration of cells. In this study, a HeLa cell wound-healing assay under different treated conditions was carried out to observe the effect of **QW12** on HeLa cell migration. As shown in Figure 3, the change in scratch closure indicated that cells treated with **QW12** significantly prevented wound healing. After 48 h, the rate of scratch in closure with different concentrations of **QW12** (10 and 40 μM) was 28.0% and 17.2%, respectively. All values in the **QW12**-treated groups were much lower than in the control group (55.7%). These data indicated that **QW12** markedly inhibited HeLa cell would-healing in a dose- and time-dependent manner, which means that **QW12** can inhibit HeLa cell migration.

To evaluate the effect of **QW12** on cell proliferation, we performed plate-cloning assays. In the plate-cloning experiment, cells treated with different concentrations of **QW12** proliferated at a significantly lower rate than the control (Figure 4B). Therefore, the results implied that **QW12** could markedly inhibit HeLa cell proliferation.

#### 3.2.3. **QW12** Induces Intracellular ROS Production

Reactive oxygen species (ROS) are generated as by-products of normal aerobic metabolism or as second messengers in various signal transduction pathways in response to oxidative stress, which can elicit a wide spectrum of biological responses, such as macromolecular damage and cell death [25,26,27]. Substantial evidence suggests that the generation of ROS is part of the mechanism by which many anticancer agents kill tumor cells [28,29,30]. It is reported that nifuroxazide could induce apoptosis through ROS accumulation [15]. Although the 5-nitro group is believed to be the main responsible mechanism for the generation of ROS for nifuroxazide, there is evidence that ROS generation can also be independent of the presence of the nitro group [31]. Since the side chain of **QW12** is similar to nifuroxazide, we were motivated to figure out whether **QW12** could also trigger ROS production in HeLa cells. As illustrated in Figure 5A,B, compound **QW12** stimulates ROS accumulation in a dose-dependent manner in HeLa cells. To further confirm the correlation between **QW12**-induced intracellular ROS accumulation and antiproliferation activity, HeLa cells were pretreated with scavenger *N*-acetylcysteine (NAC) for 1 h, then **QW12** was added, and cell viability was examined. The results showed that the NAC pretreatment of cells blocked the accumulation of ROS induced by **QW12** (Figure 5D,E), and NAC significantly blocked the antiproliferative effect of **QW12** (Figure 5C), suggesting that the elevation of ROS levels is a critical event in the proliferation inhibition induced by **QW12**.

Previous studies have reported that ROS could inhibit the PI3K/AKT pathway in various tumor cells treated with antitumor drugs [32,33]. Therefore, we attempted to explore the effect of **QW12** on this pathway in HeLa cells. As shown in Figure 5F, treatment with **QW12** significantly decreased phosphorylated AKT levels (T308), while total AKT was not affected. Furthermore, the inhibitory effect of **QW12** on AKT phosphorylation could be reversed by NAC treatment. Thus, these results confirm that **QW12** could block the PI3K/AKT pathways in a ROS-dependent fashion.

#### 3.2.4. **QW12** Induces the Apoptosis of HeLa Cells

Hoechst 33324 dyes can stain concentrated nuclei, thus distinguishing apoptotic cells from normal cells. Therefore, we used Hoechst 33,342 to stain the nucleus of HeLa cells to analyze the inhibitory effect of **QW12** on cell apoptosis. As shown in Figure 6A, significant apoptotic morphological characteristics were observed in the nucleus of cells treated with **QW12** compared to the control group, while the nucleus of control cells was rectangular in shape without any apoptotic bodies.

Annexin V-FITC is used to detect the early stage of apoptosis. PI can pass through the cell membrane to stain the cells in the middle and late stages of apoptosis. Therefore, when Annexin V-FITC is used in combination with PI, cells in different stages of apoptosis can be distinguished. As shown in Figure 6B, **QW12**-treated HeLa cells clustered mostly in the upper right quadrant. Cell apoptosis rates treated with **QW12** at concentrations of 10 μM and 40 μM are 13.9% and 34.7%, respectively. They were much higher than in the control group (3.5%) (Figure 6C). These results suggested that **QW12** has great potential application in inducing HeLa cell apoptosis.

To investigate whether the expression of pro- or anti-apoptotic proteins in the Bcl-2 family undergoes changes during **QW12** treatment, we determined the expression of Bax and Bcl-2. Bax expression increased slightly, while Bcl-2 expression decreased significantly compared to the control group (Figure 6D). Therefore, Western blot analysis (Figure 6E) revealed a significant downregulation of the Bcl-2/Bax expression ratio in the 40 μM (0.66 ± 0.01) group compared to the control group (1.00) (Figure 6E).

#### 3.2.5. **QW12** Blocks HeLa Cells in Phase S

To investigate the effect of **QW12** on the distribution of the cell cycle, a flow cytometric analysis was performed. HeLa cells were treated with 0, 10, and 40 μM of **QW12** for 24 h and then subjected to a flow cytometric analysis after DNA staining. As shown in Figure 7A, the untreated cells exhibited the expected pattern for continuously growing cells, while the cells treated with **QW12** progressively increased during the S phase of the cell cycle at 10 and 40 μM. For example, the HeLa cell population gradually increased from 23.19% at 10 μM to 51.88% at 40 μM in the S phase (Figure 7B). Simultaneously, the percentage of cells in phases G0/G1 and G2/M decreased significantly, indicating that **QW12** mainly arrests the cell cycle in phase S.

#### 3.2.6. **QW12** Inhibits STAT3 Y705 Phosphorylation in Cell-Based Assays

The activation of STAT3 is tightly regulated during various physiological processes, and the aberrant and persistent activation of STAT3 has been found in various types of cancer [34,35,36]. Nifuroxazide, which has a furan–hydrazone core similar to **QW12**, inhibited STAT3 [14,15,16]. Western blot analysis determined whether **QW12** could also regulate the STAT3 signaling pathway in HeLa cells. HeLa cells were treated with **QW12** for 12 h, and total cell lysates were prepared; Y705-phosphorylated STAT3 proteins were detected using a specific antibody. As shown in Figure 8, **QW12** inhibited Tyr705 phosphorylation in a dose-dependent manner, and the total STAT3 level also decreased at these concentrations. Therefore, these results indicated that **QW12** effectively inhibit STAT3 phosphorylation.

#### 3.2.7. Kinetic Affinity of **QW12** versus STAT3

To confirm whether **QW12** is a direct STAT3 inhibitor, a BLI (biolayer interferometry) analysis was performed. Briefly, recombinant STAT3 with a His tag was immobilized on SSA biosensors, and the dissociation constants were determined by measuring the binding to serial dilutions of **QW12** at concentrations ranging from 5.2 to 333 μM (Figure 9A). The signal was collected, and the software calculated the kinetic affinity, which showed that the steady-state curve reaches saturation (Figure 9B). The binding affinity (KD) was calculated as 67.3 μM. The results indicated that compound **QW12** has a moderate binding affinity to the STAT3 protein.

### 3.3. Compound **QW12** Docking Study with the STAT3 SH2 Domain

To account for the potent STAT3 phosphorylation-inhibition activities of **QW12** in HeLa cells, a molecular docking study was carried out to evaluate potential interactions with the STAT3 SH2 domain (PDB: 1BG1). As shown in Figure 10, the hydroxyl group in the phenyl ring of the side-chain hydrazone formed two hydrogen bonds with residues Arg595 (3.0 Å) and Lys591 (2.7 Å), and the carbonyl oxygen atom of the hydrazone moiety also interacted with residue Lys591 (3.3 Å) through hydrogen bonding. The N atom in the hydrazone group could interact with Ser636 (3.3 Å) by hydrogen bonding. The oxygen atom of the methoxy group immediately next to the furan ring formed a hydrogen bond with Gln635 (3.4 Å). On the basis of in silico results, it would be beneficial to introduce a side chain on the methoxyphenyl ring to occupy the pY subpocket, thereby increasing the selectivity and binding affinity. Moreover, a suitable conformation of **QW12** contributed a lot to the tight binding. The predicted binding energy was −56.6 kcal/mol. These docking results explained that the hydrazone moiety and the substitution of p-hydroxyl may be necessary to target the STAT3 SH2 domain to some extent.

## 4. Conclusions

In this study, we designed and synthesized a series of quinoxaline–furan derivatives and evaluated their antiproliferative effects in vitro. Among these compounds, **QW12** was found to be the most potent hybrid against five cancer cell lines, especially inhibiting HeLa cells with an IC_50_ value of 10.58 μM. **QW12** was also shown to inhibit the tumor migration and invasion of HeLa cells. Furthermore, it increased the production and accumulation of ROS in HeLa cells, which accounts for its partial antiproliferation effect through inducing cell apoptosis. The Hoechst 33342 and annexin V-FITC/PI staining experiments proved that **QW12** induced the apoptosis of HeLa cells. The protein expression of anti-apoptotic and pro-apoptotic proteins Bcl-2 and Bax was also affected by **QW12** in a concentration-dependent manner. Western blot analysis indicated that **QW12** inhibited STAT3 phosphorylation levels. The kinetic affinity assay confirmed that **QW12** could directly bind to STAT3 with a KD value of 67.3 μM. The docking study implied that the compound **QW12** bound to the cavities pY+1 and pY-X of the STAT3 SH2 domain. In summary, all of these data provided a structural reference for the development of a novel scaffold and candidate for tumor treatment.

## Data Availability

The data presented in this study are available on request from the corresponding author.

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
