# Peer review of "Discovery and SAR Study of Quinoxaline–Arylfuran Derivatives as a New Class of Antitumor Agents"

_pharmaceutics, 2022, doi:10.3390/pharmaceutics14112420_

Round 1

Reviewer 1 Report

The present manuscript describes the synthesis of quinoxaline-arylfuran hybrids and their potential antitumor activity.

In this work the authors synthesize a series consisting of sixteen compounds by joining the quinoxaline structure to a substituted furan ring, which is then further modulated through acylhydrazone formation. A series of eight naphatlene derivatives is synthesized too. The synthesized structures are original and obtained through a straightforward synthetic route.

The synthesized compounds are then tested for their cytotoxic activity in four cell lines at the single concentration of 20 microM. From the preliminary SAR drawn, compound QW12 is selected and further studied to try to understand its putative mechanism of action.

Authors discover that QW12 is able to prevent cell migration and proliferation and to increase cellular ROS production inducing apoptosis in HeLa cells. This compound is also able to inhibit STAT3 phosphorylation.

Overall, the work is well done and the experimental procedures sound. A new scaffold potentially useful for STAT3 inhibition is disclosed.

According to my opinion, there are some points which need to be smoothed before the manuscript can be accepted for publication.

The major concern, regards the rational of the work:

11)      The design of hybrids appears to be based more on synthetic feasibility rather than on rational approach. Usually, hybrid drugs (multi-target drugs) are obtained by joining two pharmacophores able to modulate targets through a known mechanism of action. This strategy allows to obtain molecules that efficiently combine two mechanism of actions to finally obtain an improved pharmacological effect, this is not the case of the present work.

22)      The rational of using furan in an antitumor drug is weak. Nifuroxazide (an antibacterial nitrofuran derivative) can exert antiproliferative activity because of its redox properties rather than because of the presence of a furan, while the mechanism of action of compounds 8 and 9 is not described. Therefore, it is difficult to understand how the authors decided to synthesize these compounds.

To strengthen the rational of the work it can be useful to better illustrate the mechanism of action of some of the reported examples rather than just giving a reference to other work.   

Other points should be checked or better explained:

11)      Why did the authors use WP1066 as the comparator? This is not mentioned in the text. WP1066 is an electrophilic JAK inhibitor, structurally different from the synthesised compounds.

22)      Is the inhibition of migration by QW12 at 24 and 48 h statistically different from each other?

33)      It is conceivable that the authors test the modulation ROS production evoked by QW12 in cells.  However, nifuroxazide, to the best of my knowledge, act as ROS inducer because it is a nitrofuran derivative not simply a furan. Do the authors have evidence for a different mechanism of ROS-induction by nifuroxazide?

Minor points to be checked:

Page 11, line 1: Gleichzeitig?

Structures of compounds QW4, QW12 and QW20 in scheme 1 are not correct: the OH group is missing.

References 3-7 all dates 2011, are there more recent examples of quinoxaline-containing biologically active compounds?

Reference for activity of compound 4 (Figure 1) is missing.

Page 2, line 15: “…and favorable rat pharmacokinetic….”, should read “ …and shows favorable rat pharmacokinetic….”.

Complete references must be given. Abbreviations such as “ibid”, are not acceptable.

Reviewer 2 Report

The manuscript  (ID: pharmaceutics-1940158) titled “Discovery and SAR of Quinoxalines-Arylfuran Hybrids as a New Class of Antitumor Agents” by Dongmei Fan, Pingxian Liu, Yunhan Jiang, Xinlian He, Lidan Zhang, Lijiao Wang, Tao Yang is adequate to publish in the Pharmaceutics journal.

Text needs some editorial correction, for example page 2 on the bottom, a few words are written in a bigger font like : “potent”, “activity”, “melanoma”, “in”. The whole text needs correction and serious attention.

In Scheme 1. Synthetic routes for compounds QW1-24,  the numbers of compounds in the text 2.1 Chemistry are completly different than in the scheme below. In the experimental part 4.1 Chemistry, the numbers of chemical compounds are the same like in the scheme, but different in the text of page 3. 

In the Experimental part for 11a and 11c the NMR spectra are not correct; the numbers of C and H are less than in the molecular formula (C16H15N2O2), (C20H21O4) respectively. For 14a, 14b, H (protons are not correct 14c C(carbon) is not correct.  All compounds QW, particularly their spectra H and C, need specific attention to check their amount compared to  molecular formulas written. 

Reviewer 3 Report

The paper deals with a very important and current group of antitumor compounds, with hybrids, however, many comments and questions should be answered. Moreover some important data are needed.

1.       Fig. 2.: What is the role of the hydrazone moiety? Why call it linker? In hybrid chemistry the linker is between the two pharmacophores. In this case the „linker” is coupled with different aromatics and heteroaromatics.

2.       Table 1.: it would be necessary to present the activity of a known anticancer agent as a control. Without it, the data are not relevant. The authors carried out extensive biological tests in most places without known controls

3.       What is the structure of WP1066? It should be presented.

4.       ROS production: What is the role of N-acetylcysteine? A cited paper would be required as Ref. for the explanation.

5.       Docking studies: Can some kind of conclusions be drawn as to what changes should be further made to the structure?

6.       Experimental: the hyphen before the name of compounds is confusing. Why is it even there?

7.       Name of compounds: E and N' are in italics as prefixes referring to structure

Round 2

Reviewer 3 Report

Based on the author's answers and the revised paper I can accept the manuscript without any changes.